# Polydopamine Nanosphere with In-Situ Loaded Gentamicin and Its Antimicrobial Activity

**DOI:** 10.3390/molecules25092090

**Published:** 2020-04-30

**Authors:** Rahila Batul, Mrinal Bhave, Peter J. Mahon, Aimin Yu

**Affiliations:** Department of Chemistry and Biotechnology, Faculty of Science, Engineering and Technology, Swinburne University of Technology, Hawthorn, VIC 3122, Australia; rbatul@swin.edu.au (R.B.); mbhave@swin.edu.au (M.B.); pmahon@swin.edu.au (P.J.M.)

**Keywords:** polydopamine, drug delivery, gentamicin, antimicrobial property

## Abstract

The mussel inspired polydopamine has acquired great relevance in the field of nanomedicines, owing to its incredible physicochemical properties. Polydopamine nanoparticles (PDA NPs) due to their low cytotoxicity, high biocompatibility and ready biodegradation have already been widely investigated in various drug delivery, chemotherapeutic, and diagnostic applications. In addition, owing to its highly reactive nature, it possesses a very high capability for loading drugs and chemotherapeutics. Therefore, the loading efficiency of PDA NPs for an antibiotic i.e., gentamicin (G) has been investigated in this work. For this purpose, an in-situ polymerization method was studied to load the drug into PDA NPs using variable drug: monomer ratios. Scanning electron microscope (SEM), Fourier-transform infrared spectroscopy (FTIR), and X-ray photoelectron spectroscopy (XPS) confirmed the successful loading of drug within PDA NPs, mainly via hydrogen bonding between the amine groups of gentamicin and the hydroxyl groups of PDA. The loading amount was quantified by liquid chromatography–mass spectrometry (LC-MS) and the highest percentage loading capacity was achieved for G-PDA prepared with drug to monomer ratio of 1:1. Moreover, the gentamicin loaded PDA NPs were tested in a preliminary antibacterial evaluation using the broth microdilution method against both Gram-(+) *Staphylococcus aureus* and Gram-(−) *Pseudomonas aeruginosa* microorganisms. The highest loaded G-PDA sample exhibited the lowest minimum inhibitory concentration and minimum bactericidal concentration values. The developed gentamicin loaded PDA is very promising for long term drug release and treating various microbial infections.

## 1. Introduction

Nanomedicines have played a surprising role in the biomedical field in terms of various physical and chemical responses for drug delivery [1,2]. A considerable number of nanocarriers have been investigated for drug delivery including organic and inorganic nanocarriers for targeted drug delivery and chemotherapy. These nanocarriers help in improvement in drug cellular accumulation and reduction of the required dosing frequency, hence improving patient compliance and therapeutic efficacy [3,4].

In the last decade, polydopamine (PDA) materials have received tremendous attention and have been applied to various fields, including drug delivery [5,6]. Before its nanoparticulate formulation, PDA was come into considerations as a modifier and/or coating material because of its robust adhesive nature in weakly alkaline conditions [7]. Both above-mentioned formulations rely on a single step self-polymerization process of the biological molecule, dopamine. Dopamine was derived from catechol-containing compounds lysine and 3,4-dihydroxyphenylalanine, located in the marine mussel’s foot, responsible for their adhesion even on wet surfaces. As dopamine can efficiently bond with nucleophilic species, such as amine and thiol functional groups via Michael-addition or Schiff-base reactions, likewise, PDA can be feasibly grafted in the form of a film, with various functional moieties in a process that is substrate-independent and has high chemical reactivity [8]. Messersmith’s group has listed a group of substrates that were successfully coated with PDA in their recent review [9]. Based on a surface independent coating process, ease of deposition and provision of very reactive functional groups for further modification, decades of PDA-based nanocomposites have been reported so far [10,11]. These discoveries have revealed the various chemical and physical properties of PDA such as metal chelation, reducing agent [12,13], optoelectrical properties [14], chemo-phototherapeutic agent [15,16], photoacoustic agent [17], and biosensor [18] and other mechanical properties [19]. In addition, the implementation of polydopamine nanoparticles (PDA NPs) and other biomimetic core-shell nanostructures in the field of drug delivery and tissue engineering is a very hot area [10,20].

Owing to its superior properties such as low toxicity, high biocompatibility, and biodegradation, and the cheap and convenient procedure to produce PDA NPs, herein, we have investigated their implementation in antibiotic loading. The main methods for drug loading using PDA nanocarriers that have been reported so far are: physical adsorption of drugs with a terpenoid structure, chemical bonding of the drug to the outer surface of PDA using chemical linkers, and formation of dopamine-drug conjugates [21]. There have been very few reports on drug loading into dopamine via in situ polymerization method [22,23]. Instead, post-loading was more commonly used which involves a two-step process: Synthesis of NPs; and drug loading into as-prepared NPs [24,25]. In some work, polymer-drug conjugates were designed to improve the therapeutic efficacy of the drug by controlling the drug release [26]. However, this method requires an additional drug conjugation step prior to loading. To save labor and time, we propose a facile one step method for drug loading, i.e., synthesis of PDA NPs with drug loading carried out simultaneously. This method only requires atmospheric oxygen and alkaline conditions for dopamine polymerization to start with and simultaneous drug loading, hence, excluding the fussy chemical grafting process and the use of organic solvents, which also makes it an environment friendly method to synthesize drug-loaded nanocarriers. Moreover, this method also provided higher drug loading capacity as compared to post loading method used by other researchers [27]. Such an effective, low cost and simple method may have great potential in controlled drug delivery applications.

Gentamicin, an aminoglycoside antibiotic, has been commonly used to treat many types of bacterial infections such as respiratory and urinary tract, blood, bone and soft tissue infections, caused by Gram-negative bacteria including *Pseudomonas*
*aeruginosa*, *Proteus* spp., *Escherichia coli, Klebsiella pneumoniae, Enterobacter aerogenes, Serratia* spp., and the Gram-positive *Staphylococcus* [28]. This drug attaches to the bacterial cells via a 30S ribosomal subunit, thus causing misreading of the genetic code and inhibits translocation [29]. Many clinicians have minimized its use because of its shorter half-life, low bioavailability, and serious side effects, which can be caused even by using a single dose regimen, such as ototoxicity and nephrotoxicity [30]. Moreover, the low intestinal absorption of gentamicin, like other aminoglycosides, also limits its administrative methods to intravenous, intramuscular, and topical to get rid of infections [31,32]. To improve the therapeutic efficacy and to minimize the side-effects associated with gentamicin, a nanoformulation approach with targeted drug delivery might be beneficial. Loading of gentamicin has already been investigated by using various nanoparticles such as niosomes [33], poly(lactide-co-glycolide) (PLGA) [34], chitosan [35], poly(3-hydroxybutyrate) [36], calcium carbonate [37], gold nanoparticles [38], and mesoporous silica [39].

Here we report the development of gentamicin loaded PDA NPs by an in-situ one-step polymerization method. This method is simple and has high drug loading efficiency with retained antimicrobial activity. The as-synthesized NPs are characterized by various techniques including scanning electron microscope (SEM), Fourier-transform infrared spectroscopy (FTIR), and X-ray photoelectron spectroscopy (XPS). The drug loading quantitation is conducted using liquid chromatography–mass spectrometry (LC-MS). Additionally, the antibacterial effects of gentamicin loaded PDA NPs against commonly causing infectious organisms such as *Staphylococcus*
*aureus* and *Pseudomonas aeruginosa* are assessed as well.

## 2. Results and Discussion

### 2.1. Synthesis and Characterization of PDA and G-PDA NPs

PDA NPs were synthesized through the oxidative self-polymerization of dopamine for 24 h in a mixed solution of water, ammonia, and ethanol at pH 12.5. Although it is still not fully understood, the polymerization mechanism of PDA involves the oxidation of dopamine, triggered by alkaline pH condition, transforming dopamine into dopamine quinone. The dopamine quinone then undergoes an irreversible intramolecular cyclization through the 1,4 Michael addition, which converts dopamine quinone into leukodopaminechrome. Thereafter, the obtained oxidative product leukodopaminechrome undergoes further oxidation and rearrangement to form 5,6 dihydroxyindole. Afterward, this 5,6-dihydroxyindole leads to the product formation via two pathways: Covalent oxidative polymerization and non-covalent self-assembly, as hypothesized by Hong and co-workers [40].

Gentamicin (G) was chosen as a model drug to be in-situ loaded into PDA NPs during the polymerization process of dopamine. This was simply achieved by adding gentamicin into the dopamine solution. A series of G-PDA NPs were synthesized by varying the amount of gentamicin and keeping the amount of dopamine constant.

To get an insight of the gentamicin loading into PDA, FTIR spectra were obtained as shown in Figure 1a. For the PDA sample, the peak at 3233 cm^−1^ corresponds to the stretching vibration of O–H of PDA, whereas the band at 1573 cm^−1^ is attributed to the stretching and bending vibration of N–H and the characteristic C=C bonds from the indole structure of PDA was proven by peaks at 1558 and 1506 cm^−^^1^. Moreover, a small peak at 1281 cm^−1^ corresponds to the stretching vibration of catechol hydroxyl C–O and/or C–N [41]. The presence of indole and indolequinone structures in the FTIR spectrum of PDA proved the successful polymerization of dopamine to PDA [24]. In the FTIR spectrum of gentamicin sulphate, the peaks at 1621 and 1526 cm^–1^ correspond to the amide (N–H) bending vibrations of primary aromatic amines. The peaks at 605 and 1035 cm^–1^ are attributed to the sulfur content in the form of a S–O bending vibration and S–O stretch, respectively. In the FTIR spectrum of G-PDA NPs, some additional peaks were observed. The peak at 1337 cm^−1^ corresponds to C–N stretching of gentamicin [42]. Moreover, the peaks at 1526 and 1035 cm^−1^ corresponding to N-H and S–O groups in gentamicin spectrum shift to 1537 and 1055 cm^−1^, respectively after loading into PDA NPs [43]. Therefore, the presence of additional gentamicin peaks in the FTIR spectrum of G-PDA NPs can provide evidence of successful loading of gentamicin into PDA NPs. However, the presence of all PDA peaks after in situ gentamicin loading but with lower intensities might indicate possible chemical interaction between amino groups of gentamicin and hydroxyl groups of PDA. Therefore, to further investigate the chemical interaction mechanism, XPS analysis was carried out.

XPS survey spectra were used to investigate the surface chemical composition of NPs before and after gentamicin loading, as shown in Figure 1b. The atomic percentages are shown in Table 1. The percentage changes of the main atoms including C1s, N1s, O1s, and S2p were observed after drug incorporation. The nitrogen content increased from 7.99% to 8.51%, while the carbon content decreased from 73.58% to 62.33%. Correspondingly, the ratio of N/C increased from 0.11 to 0.14. Owing to the nitrogen rich nature of gentamicin, an increase in nitrogen content confirmed the successful loading of gentamicin in PDA NPs. Notably, a sulfur contamination was observed in PDA NPs, which might come from the adjacent sample and/or by repetitive XPS scans, which would be considered as an experimental error [44]. However, an increase in sulfur content was observed from 0.28% to 1.97% after gentamicin loading, which was expectedly came from gentamicin after loading.

The high resolution of XPS spectra of C1s, N1s and O1s allowed peak fitting to be performed to investigate the mechanism underlying the formation of gentamicin loaded PDA NPs. The C1s spectrum of PDA sample could be curve fitted into four components with binding energies 284.0, 285.6, 287.2, 288.5, and 290.4 which are assigned to C–C, C–N, C–O, O=C and π-π*, respectively as shown in Figure 2a. After gentamicin loading, the π-π* satellite in PDA NPs disappeared confirming the drug entrapment in PDA NPs. Moreover, the increase in relative intensities corresponding to C-N and C-O was observed after gentamicin loading as shown in Figure 2d. Furthermore, the high resolution N1s spectra of PDA was resolved into three components namely, C–N, C–N–C, and =N–C as shown in Figure 2b. After gentamicin loading, the relative intensities for C–N was also increased from 35.24% to 42.54%, as shown in Figure 2e. Additionally, the high resolution O1s spectra (Figure 2f) also showed an increase in the ratio of absorptive peak areas for O–C bond after drug loading. The atomic percentages with corresponding chemical groups for each peak within the enveloped region are presented in Appendix A. At our working pH (12.5), most of the amino groups in gentamicin are neutral, thus they have more tendency to form hydrogen bonds with hydroxyl groups of PDA. Overall, XPS data of G-PDA NPs concludes that the increase relative intensities as well as concentrations of C–N and O–C bonds in N1s and O1s spectra respectively, supports the presence of hydrogen bonding, responsible for gentamicin loading in PDA NPs [45]. The possible mechanism of interaction between PDA and gentamicin is schematically shown in Figure 3.

The successful loading of gentamicin was further confirmed by the surface charge change of the PDA particles. The zeta potential of PDA NPs was measured to be −31 mV in water which was attributed to the deprotonation of phenolic hydroxyl groups of PDA at neutral pH [46]. The zeta potential of PDA NPs was shifted towards being less negative when gentamicin was loaded. This is owing to the presence of more positively charged amine groups in gentamicin that neutralizes the negative potential of PDA, as shown in Table 2. Furthermore, when an equal ratio of gentamicin and dopamine (1:1), the zeta potential of G-PDA NPs become near to zero. This observation confirmed that the surface of PDA NPs was entirely covered with gentamicin via adsorption of drug onto the PDA NPs.

The morphology of PDA and G-PDA NPs was observed by SEM, as shown in Figure 4. The PDA NPs are spherical and uniform with an average diameter of 165 nm, as shown in Figure 4a. After gentamicin loading, the sizes of G-PDA particles increase significantly. Additionally, the particle size increases (from 250 to 550 nm) with increasing drug ratio in dopamine solution, (Figure 4b–e). This size increment after drug loading supports the successful gentamicin loading into PDA NPs. The hydrodynamic particle size that were measured by the particle size analyzer, as shown in Table 2, are consistent with SEM results.

### 2.2. Quantitation of the Amount of Gentamicin Loading

Gentamicin is highly polar in nature due to the presence of a large number of hydroxyl and amino groups in its structure, which causes its poor retention when using conventional reversed phase high-performance liquid chromatography (HPLC). Therefore, hydrophilic interaction liquid chromatography (HILIC) was used to analyze the loaded amount of gentamicin in PDA NPs [47]. Gentamicin and its three stereoisomers were successfully retained and identified based on their specific *m*/*z* values viz; C1: 478 *m*/*z*, 5.8 min, C1a: 450.2 *m*/*z*, 5.4 min, C2/C2a/C2b: 464.3 *m*/*z*, 5.5 min. A standard curve for gentamicin was obtained for the concentration range of 0.5 to 5 mg/mL, which showed a very good linearity giving a R^2^ value of 0.996 as shown in Figure 5a. Notably, the peak areas of all the isomers were summed up to obtain the calibration curve as well as for the quantitation of drug loading.

Based on the calibration curve, the mass percentage of drug loaded inside the particles was calculated, as shown in Table 2. An increase in initial gentamicin concentration with respect to dopamine resulted in higher loading capacity. By increasing the ratio of gentamicin in the polymerization solution from 0.4:1 to 1:1, the initial amount of gentamicin increased by 2.5 fold, which resulted in the increased loading percentage of gentamicin within PDA NPs from 66.6% to 82.2%. However, when the ratio of gentamicin to dopamine was high, the loading percentage increase became less, and the maximum drug loading was achieved with the highest gentamicin to dopamine ratio i.e., G-PDA 1:1, as shown in Figure 5b. These findings suggest a very high loading efficiency of PDA that proved it to be a very efficient nanoreservior for gentamicin.

### 2.3. Antimicrobial Activity of G-PDA NPs

In order to investigate the antimicrobial effect of gentamicin loaded PDA NPs, the broth micro dilution method was performed against both Gram-positive and Gram-negative bacteria. For this purpose, minimum inhibitory concentration (MIC) and minimum bactericidal concentration (MBC) values were assessed with the aim to evaluate whether the gentamicin maintains its activity after in-situ loading. After 24 h incubation, the well with no visible growth was taken as MIC. The turbid wells could be clearly observed for the presence of bacteria (Appendix A). Unexpectedly, the MIC of gentamicin against both bacterial species were observed to be same at the concentration of 1.95 µg/mL. PDA NPs did not show any clear well, whereas MIC of G-PDA NPs was observed at the concentration of 0.6 mg/mL. Then the samples from the clear wells were cultured on nutrient agar plates. After 24 h, visible colonies on the plates were observed for *S. aureus* as shown in Figure 6a–d.

However, MBC values against *P. aeruginosa* were determined by the color change of the culture medium as shown in Figure 6e–h. This was because of pyocyanin secretion by *P. aeruginosa* species, causing diffusion of colonies throughout the media [48]. MBC for gentamicin was obtained as 3.9 µg/mL whereas MICs and MBCs of drug-loaded nanoparticles with variable gentamicin to dopamine ratios are presented in Table 3.

From above findings, we could conclude that G-PDA NPs exhibit an antimicrobial effect against the treated microorganisms. Additionally, the antimicrobial effect increased when the gentamicin loading amount in PDA NPs increased, as indicated by MIC and MBC values for G-PDA 1:1, which were comparatively lower than other batches with less gentamicin loaded. Notably, these MIC values are based on drug that is released into the solution, highlighting that PDA have negligible antimicrobial activity. The higher MIC values as compared to free drug might be because some drug molecules are still entrapped inside the particles. It is therefore necessary for this to be released using other possible release mechanisms. Work is underway in our laboratory to study various mechanisms for improving gentamicin release.

## 3. Materials and Methods

### 3.1. Materials

Dopamine hydrochloride, ammonium hydroxide (38–40%) and ethanol were purchased from Sigma-Aldrich (Pty. Ltd., Sydney, NSW, Australia). Gentamicin sulphate was purchased from AK Scientific (Union city, CA, USA). Chromatography grade acetonitrile, ammonium formate and formic acid were purchased from Sigma-Aldrich. All the solutions used in the following experiments were prepared from Milli Q water (Millipore Simplicity system) having a resistivity of 18.2 MΩ cm. Nutrient agar and Nutrient broth were supplied by Acumedia (Melbourne, Australia). The microbial species tested were *S. aureus* (ATCC 25923) and *P. aeruginosa* (ATCC 9721).

### 3.2. Synthesis of PDA NPs

PDA NPs were synthesized by a procedure similar to previous reports [5,49]. Briefly, a solution mixture of 10 mL of ethanol, 0.5 mL of aqueous ammonia solution and 22.5 mL of water was kept under stirring for 30 min. Then 0.125 g of dopamine previously dissolved in 2.5 mL water was injected into the above water–ethanol mixture. The reaction was allowed to proceed under stirring for 24 h at room temperature. Product formation was accompanied by colour change of the solution from colorless to pale yellow and then brownish-black with the passage of time. After 24 h, particles were obtained by centrifugation at 15,000 rpm for 20 min followed by three times washing with water and drying in an oven at 80 °C for 4 h.

### 3.3. Synthesis of G-PDA NPs

The gentamicin loaded PDA NPs were prepared via a similar procedure except adding certain amounts of gentamicin sulfate. The weight ratios (in mg) of gentamicin and dopamine used were 0:1 (0:125 mg), 0.4:1 (50 mg:125 mg), 0.6:1 (75 mg:125 mg), 0.8:1 (100 mg:125 mg), and 1:1 (125 mg:125 mg). Afterward, G-PDA NPs were obtained by centrifugation at 15,000 rpm for 20 min followed by three times washing with water and drying in an oven at 80 °C for 4 h. The particle yield was calculated by using the following formula:$$\%\;\mathrm{Yield}=\frac{\mathrm{Mass}\;\mathrm{of}\;\mathrm{particles}\;\mathrm{obtained}}{\mathrm{Total}\;\mathrm{weight}\;\mathrm{of}\;\mathrm{dopamine}\;\mathrm{and}\;\mathrm{drug}\;\mathrm{used}}\times 100$$

### 3.4. *Determination of Gentamicin Loading in PDA NPs*

All the supernatents were filtered through a 0.22-micron filter medium, and their pH was adjusted to 5.5 to fulfil the column pH conditions. The samples were centrifuged at 15,000 rpm for 10 min and the light brown supernatants were collected to be analyzed for LC-MS. Hydrophilic interaction liquid chromatography (HILIC) chromatography method was adopted as proposed by Kumar et al. [47]. The apparatus used for this purpose was Agilent 1290 Infinity II UHPLC with a 6545 Q-TOF MS (Agilent Headquarters, Santa Clara, CA, USA). Data acquisition and analysis were carried out by Mass Hunter B.08.00 software. The separation was carried out at 40 °C by zwitterionic (ZIC^®^˗HILIC, 2.1–150 mm, 3.5 µm, SeQuant AB) purchased from Merck (Melbourne, VIC, Australia). MS detection was performed in positive ionisation mode by using a dual ESI and AJS (Agilent Jet Stream Technology) source (Agilent Headquarters, Santa Clara, CA, USA); the ion *m*/*z* values were 478.32, 468.31, and 450.29 for various gentamicin isomers viz; C1, C2/C2a/C2b and C1a, respectively. The chosen flow rate was 0.3 mL/min and capillary voltage was 3950 V. Sheath gas temperature was 350 °C and the nebulizer drying gas (nitrogen) flow rate and pressure was 10.0 L/min and 20 psi respectively. The mobile phase used for separation was 175 mM ammonium formate at pH 4.5 and 0.1% formic acid in acetonitrile. Sample injection volume was 2 µL.

Areas obtained under curves for all the isomers were summed up to determine the amount of gentamicin left in the supernatents. The amount of gentamicin was subtracted from the initial amount taken prior to loading. The percentage drug loading capacity was obtained by the following formula:$$\%\;\mathrm{Drug}\;\mathrm{loading}=\frac{\mathrm{Weight}\;\mathrm{of}\;\mathrm{drug}\;\mathrm{loaded}}{\mathrm{Total}\;\mathrm{weight}\;\mathrm{of}\;\mathrm{particles}\;\mathrm{obtained}}\;\times 100$$

### 3.5. Antimicrobial Activity of Gentamicin Loaded PDA NPs (G-PDA NPs)

Antimicrobial activity of G-PDA NPs was assayed by the broth microdilution method (Appendix A) following Clinical and laboratory standards institute (CLSI) guidelines [50]. For this, twelve two-fold serial dilutions of free drug (1000 µg/mL) and G-PDA NPs (5 mg/mL) were prepared. PDA NPs (5 mg/mL) with no drug was used as the control. The drug-loaded NPs were kept in sterile distilled water for 4 to 5 days prior to release an adequate amount of drug in order to test its antimicrobial effect. A Gram-(+) *Staphylococcus aureus* (*S. aureus*) ATCC25923 and a Gram-(−) *Pseudomonas aeruginosa* (*P. aeruginosa*) (ATCC9721) were chosen to evaluate the antimicrobial activity of G-PDA NPs. These were grown in nutrient broth (NB) for 24 h at 37 °C. Actively growing inoculum was diluted with NB to an optical density of 0.1 (A600) and then further diluted to give a starting inoculum of 108 colony-forming units (CFU)/mL. Diluted *S. aureus* and *P. aeruginosa* (100 μL) was added to each well of a sterile 96-well plate. 25 μL from each of the intended samples were added to each well already containing the bacterial inoculum. The plates were prepared in triplicate. These plates were incubated for 24 h in 37 °C shaker incubator at 250 rpm. The lowest concentration of the sample at which no observable growth was apparent after 24 h was designated as the minimum inhibitory concentration (MIC). The minimum bactericidal concentration (MBC) was determined by the absence of bacterial growth on nutrient agar plates from 100 μL samples from the clear wells after 24 h.

### 3.6. Characterization

The hydrodynamic size measurement and zeta potential of PDA and G-PDA NPs were measured by 90 plus particle size analyzer. For this measurement, about 2 mg NPs were dispersed in 40 mL of water and sonicated before measurement. The size and morphology of PDA and G-PDA NPs were determined by scanning electron microscopy using FeSEM, ZIESS SUPRA 40VP (Jena, Germany) at an acceleration voltage of 3 kV. The dispersed particles were dried on cleaned glass slides followed by gold sputtering coating for 40s prior to performing SEM.

The changes in the chemical composition of PDA NPs before and after drug loading were analyzed by performing Fourier Transform Infrared spectroscopy (FTIR) analysis using the Thermo Scientific Nicolet i5 spectrometer (Waltham, MA, USA) with an iD5 ATR accessory.

To study the chemical interaction between PDA and gentamicin, X-ray photoelectron spectroscopy (XPS) was performed using AXIS Nova spectrometer (Kratos Analytical Ltd., Manchester, UK). The instrument was equipped with a monochromatic aluminum source (Al Kα, 1486.6 eV) operating at a power of 150 W which was used to determine the surface chemistry of all samples in two normal and AR conditions. The samples were prepared on 1 cm^2^ cleaned glass slides by putting a drop of dispersed particles (2 mg/mL in water) and then dried overnight. Three spots on each sample were analyzed. All elements were determined by performing survey spectra (acquired at a pass energy of 160 eV). To determine the chemical bonding, high-resolution spectroscopy was performed on elements at a detector pass energy of 20 eV. Peak fitting and data analysis was carried out by using CASA XPS software version 2.3.15. All peaks were fitted with Gaussian-broadened Lorentzian functions with a typical ratio of 70:30 using Shirley-type background. The binding energies were calibrated by setting the C1s peak at 284.8 eV and equal full-width at half-maximum were applied for all components.

## 4. Conclusions

It has been concluded that gentamicin can be loaded into PDA NPs by a simple one-step in-situ polymerization method via hydrogen bonding between the amine moieties and the hydroxyl groups of gentamicin and PDA, respectively. Moreover, by increasing the concentration of gentamicin with respect to dopamine, the loading efficiency could be increased up to 82% at an equivalent gentamicin: dopamine ratio. In other words, the PDA appeared as a very spacious nanoreservoir for gentamicin. Furthermore, the antibacterial efficiency also increased by increasing the gentamicin loading amount into PDA NPs, as revealed by MIC and MBC values for G-PDA 1:1. It was observed that some gentamicin was strongly entrapped within PDA NPs. Further investigation on developing various drug release methods is necessary to make them applicable as controlled drug delivery system for long term release of gentamicin and their efficacy against various microbial infectious agents.

## Figures and Tables

**Figure 1 molecules-25-02090-f001:**
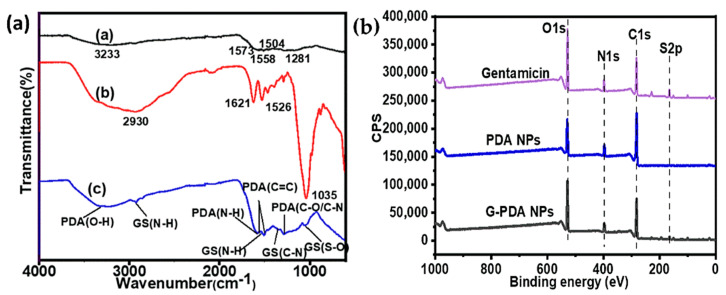
(**a**) Fourier-transform infrared spectroscopy (FTIR), and (**b**) X-ray photoelectron spectroscopy (XPS) survey spectra of polydopamine (PDA), gentamicin sulphate, and gentamicin-polydopamine nanoparticles (G-PDA NPs).

**Figure 2 molecules-25-02090-f002:**
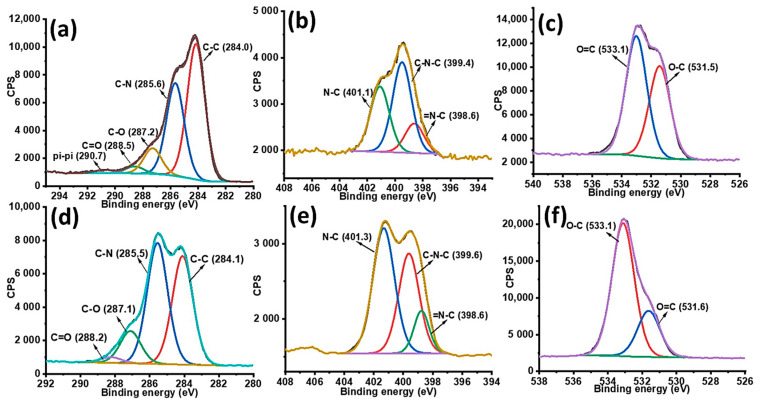
High resolution XPS spectra (**a**) C1s, (**b**) N1s; and (**c**) O1s spectra of PDA NPs; and (**d**) C1s, (**e**) N1s; and (**f**) O1s of G-PDA NPs.

**Figure 3 molecules-25-02090-f003:**
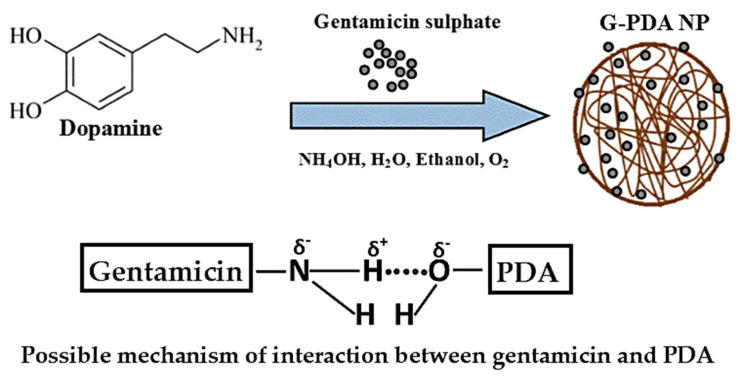
Schematic diagram showing proposed mechanism of hydrogen bonding between hydroxyl and amine moieties of PDA and gentamicin, respectively.

**Figure 4 molecules-25-02090-f004:**
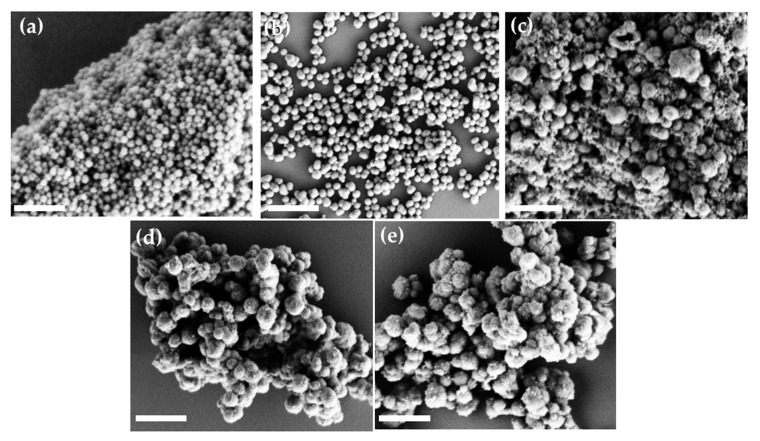
Scanning electron microscope (SEM) images of various batches prepared by changing the concentration of gentamicin; (**a**)PDA NPs; (**b**) G-PDA NPs 0.4:1; (**c**) G-PDA NPs 0.6:1; (**d**) G-PDA NPs 0.8:1; (**e**) G-PDA NPs 1:1. The scale bar is 1 µm.

**Figure 5 molecules-25-02090-f005:**
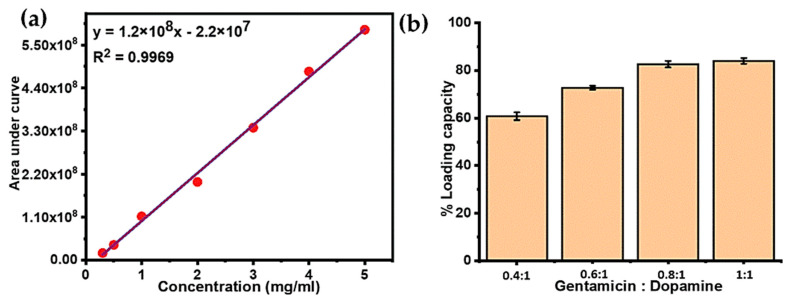
(**a**) Liquid chromatography–mass spectrometry (LC-MS) calibration curve for various gentamicin concentrations. (**b**) Mass percentage of gentamicin loading into PDA NPs with variable gentamicin to dopamine ratios.

**Figure 6 molecules-25-02090-f006:**
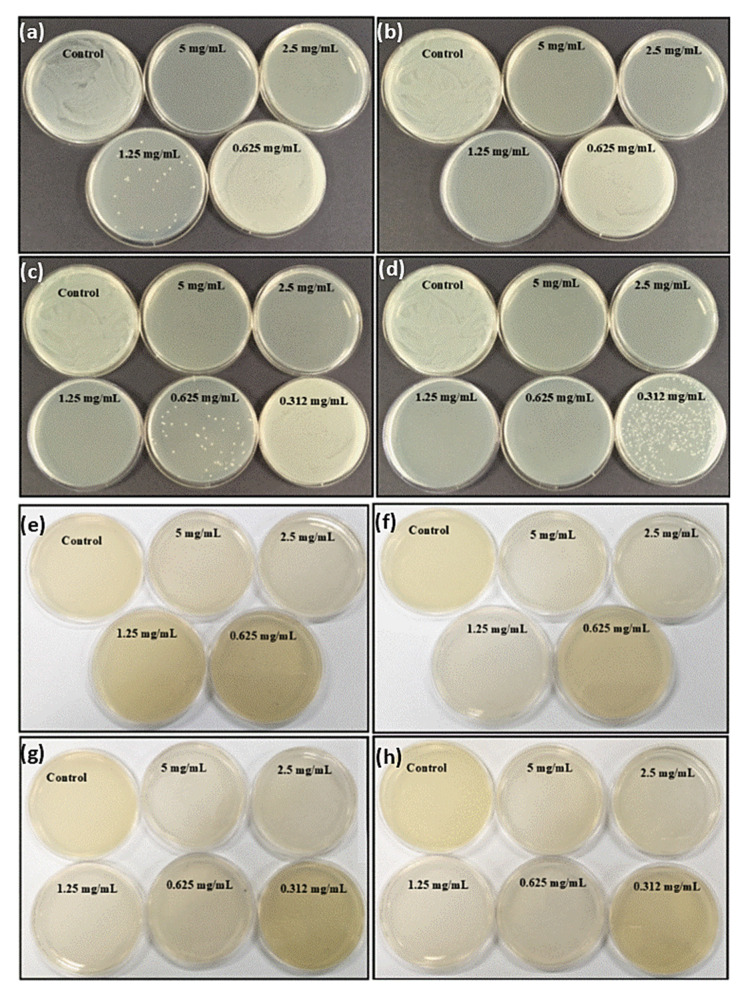
Evidence of antimicrobial activity of all batches of G-PDA NPs viz; 0.4:1, 0.6:1, 0.8:1 and 1:1 represented by figure (**a**–**d**), respectively against *Staphylococcus aureus* and (**e**–**h**) respectively against *Pseudomonas aeruginosa*. The petri dishes with no bacterial colonies representing the minimum bactericidal concentration (MBC) values for each G-PDA NPs batch.

**Table 1 molecules-25-02090-t001:** Surface elemental composition of gentamicin, polydopamine nanoparticles (PDA NPs), and gentamicin-polydopamine nanoparticles (G-PDA NPs).

Samples	Atomic Percentages				N/C	O/C
	C1s (%)	N1s (%)	O1s (%)	S2p (%)		
Gentamicin	54.96	8.54	30.62	5.88	0.16	0.55
PDA NPs	73.58	7.99	18.15	0.28	0.11	0.25
G-PDA NPs	62.33	8.51	27.18	1.97	1.14	0.44

**Table 2 molecules-25-02090-t002:** Representing particles size and zeta potential values and % loading capacity with respect to various batches.

Batch	Hydrodynamic Particle Size (nm)	Zeta Potential (mV)	Particle Yield (%)	Percentage of Loading (%)
PDA NPs	165 ± 10	−31.0 ± 0.5	11.3 ± 0.9	----
G-PDA NPs 0.4:1	292 ± 13	−22.5 ± 0.7	20.3 ± 0.9	60.8 ± 1.6
G-PDA NPs 0.6:1	375 ± 7	−13.7 ± 0.8	23.5 ± 1.8	72.7 ± 0.9
G-PDA NPs 0.8:1	449 ± 6	−6.0 ± 0.5	32.9 ± 1.4	82.7 ± 1.4
G-PDA NPs 1:1	579 ± 8	0.9 ± 0.2	33.3 ± 2.9	84.1 ± 1.2

**Table 3 molecules-25-02090-t003:** Representing the values of minimum inhibitory concentrations (MICs) and minimum bactericidal concentrations (MBCs) of various batches against *S. aureus* and *P. aeruginosa.*

Bacteria Tested	Batches	Minimum Inhibitory Concentration (MICs) mg/mL	Minimum Bactericidal Concentrations (MBCs) mg/mL
*Staphylococcus aureus* (ATCC 25923)	Gentamicin sulphate	1.95 µg/mL	3.9 µg/mL
	G-PDA NPs 0.4:1	0.625	2.5
	G-PDA NPs 0.6:1	0.625	1.25
	G-PDA NPs 0.8:1	0.312	1.25
	G-PDA NPs 1:1	0.312	0.625
*Pseudomonas aeruginosa* (ATCC 9721)	Gentamicin sulphate	1.95 µg/mL	3.9 µg/mL
	G-PDA NPs 0.4:1	0.625	2.5
	G-PDA NPs 0.6:1	0.625	1.25
	G-PDA NPs 0.8:1	0.312	0.625
	G-PDA NPs 1:1	0.312	0.625

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
