# Peer review of "Polydopamine Nanosphere with In-Situ Loaded Gentamicin and Its Antimicrobial Activity"

_molecules, 2020, doi:10.3390/molecules25092090_

Round 1
Reviewer 1 Report
The authors report about the synthesis, characterization, drug loading and antibacterial activity of PDA NP loaded with gentamicin.
The work is well written, the methodology is sound and well described. This paper seems the prosecution of their previous work "Batul, R.; Yu, A.; Bhave, M.; Khaliq, A., Synthesis of Polydopamine Nanoparticles for Drug Delivery Applications Microscopy and Microanalysis 2018, 24 1758-1759" where they already described PDA NPs synthesis and characterization.
The novelty relies then on drug loading and antibacterial activity, but as the author pointed out in the introduction, PDA and PDA NPs are a well known and versatile loading platform in theranostic. PDA NPs loading with different drugs has been already reported, although not gentamicin to the best of my knowledge.
If the Editor finds its novelty suitable for Molecules, I consider this a nice and round piece of research that could be published after minor revision.
- The authors described gentamicin loading, but not its release and proceed with a first antibacterial evaluation. Could they provide data about or discuss drug delivery?
- Figure 4: I cannot find the value of the white reference bar
- Check the reference for format mistakes (eg reference [49])
Author Response
- The authors described gentamicin loading, but not its release and proceed with a first antibacterial evaluation. Could they provide data about or discuss drug delivery?
Response: We have mentioned in the conclusion part of the manuscript regarding the ongoing work based on drug release mechanisms. The continuation of this study would be published in future.
- Figure 4: I cannot find the value of the white reference bar
Response: Thank you for your correction. It has been added.
- Check the reference for format mistakes (e.g reference [49])
Response: Correction has been made in the manuscript. All other references have also been formatted accordingly.
Reviewer 2 Report
My qualification for reviewing this manuscript is that my students and I carry out published work in this field. My view of this manuscript is that, while I am in general agreement with the conclusions of the authors, I believe that improvements can be made:
- Line 20: the monomer ratio of 5:5 bothers me; I shall return to it later.
- Line 49: I was surprised to discover that, as written by the authors, PDA is superparamagnetic (it certainly is not, since superparamagnetic behavior requires ferri- or ferromagnetic domains). Further, when I went to reference 14, I found that, while the pagination was correct, the list of authors and the title were not. I have not bothered checking whether any of the other references are incorrect, but I suggest that the authors do so.
- Line 131 and line 136: the sulfur content of pure PDA is given as 0.28%. Clearly, this is a contaminant, yet the authors make no mention of that fact. Unfortunately, such nanoscale contamination is a common occurrence, as noted, for example, in R. França et al., Journal of Colloid and Interface Science, 389 (2013) 292.
- Line 142 and elsewhere: the authors compare counts/second from different samples. This is dangerous since the number of counts/second is sample-dependent. That is, the number of counts/second will be different between two samples of the same material. It is best to ratio peak areas in a given sample.
- Line 156: the separation of peaks contained in a given spectrum should use the same full width at half maximum for all the peaks in that spectrum. For example, the O1s spectrum, in my instrument, is deconvoluted with all component peaks having widths of 1.8 eV. The same holds true for the C1s, N1s and S2p3/2 spectra, albeit with different widths. This will change the percentages in Table 1. Should the use of constant width values not fit the spectrum when the expected number of peaks is used, this does not mean that the use of constant widths is wrong; rather, it means that additional peaks are present.
- Line 172: the authors persist in using the value of 5 as the standard PDA concentration, ratioing the gentamicin concentration to it in column 1 of the table. Frankly, I have never encountered something like this before. I suggest that the numbers be ratioed, as is commonly done. Thus, 2:5 becomes 1:2.5, 3:5 becomes 1:1.67, …., and 5:5 becomes 1:1.
- Line 172: how was the particle yield (column 4) determined?
- Line 241: the compound is ammonium hydroxide.
- Line 307: FTIR stands for Fourier Transform Infrared spectroscopy.
- Line 312: the Al source is written as Al Kα, 1486.6 eV.
- Line 321: The energy in an XPS experiment are calibrated by placing the C-H/C-C peak at 284.8 or 285.0 eV. Doing so will permit a more direct comparison with the data of others.
I believe that my suggestions will enhance the manuscript.
Round 2
Reviewer 2 Report
I am basically dissatisfied with the responses of the authors, as well as with the minimal extent of their revision. Perhaps this is due to a lack of understanding of my comments, so I shall reiterate. The responses I find to be inadequate are referenced to the lines I mentioned in my original comments.
- Line 131 and line 136: had the authors read the paper I referenced, and made an effort to understand it, they would have found that minor contaminants, which would have been disregarded at the macroscale, become quite noticeable at the nanoscale. Their response, that “a minute sulphur contamination was observed in PDA NPs, which is [a] common occurrence in XPS analysis”, is simply incorrect.
- Line 142 and elsewhere: anyone with experience in analytical spectroscopic methods knows that the absolute intensity values obtained on a sample depend on the sample thickness, surface roughness, etc. This is why researchers experienced in such methods never compare absolute values between samples. Instead, they compare component peak ratios in a given sample. For the authors to defend their comparison of absolute values between samples, by saying that others have also done so, seems, to me, to indicate that they do not understand the potential seriousness of what they have done.
- Line 156: when a spectrum is analyzed, by using peaks of varying widths, that person is, in essence, fitting the spectrum with peaks at presupposed positions. That is, a bias is being imposed. The authors respond that they “haven’t observed any additional peak at constant fwhm values.”. I find this difficult to accept, for the following reason: by my rough estimation, the fwhm values in Figure 2(a) vary from about 1.5 eV for C-C, to about 2.5 eV for C-O/C-N. Had the authors fixed the fwhm value at about 1.5 eV, which is what is used in my lab, the three peaks they used could not possibly have been enough to fill the spectrum.
- Line 172: the response to how the particle yield was calculated should have been included in the manuscript.
- Line 321: the authors indicate that the energy calibration was carried out by placing the C1s C-C peak at 284.8 eV. However, Figure 2(a) has it at 284.0 eV, Figure 2(b) has it at 284.7 eV, and Figure 2(c) has it at 284.6 eV.
I believe that the authors have much more to do before their revision is found acceptable.
Author Response
I am basically dissatisfied with the responses of the authors, as well as with the minimal extent of their revision. Perhaps this is due to a lack of understanding of my comments, so I shall reiterate. The responses I find to be inadequate are referenced to the lines I mentioned in my original comments.
Response: We do apologize for our misunderstanding. We do appreciate for the reviewer’s valuable time for reviewing our manuscript thoroughly and giving us valuable suggestions. We have considered carefully all of their comments. Here are our responses below:
Line 131 and line 136: had the authors read the paper I referenced, and made an effort to understand it, they would have found that minor contaminants, which would have been disregarded at the macroscale, become quite noticeable at the nanoscale. Their response, that “a minute sulphur contamination was observed in PDA NPs, which is [a] common occurrence in XPS analysis”, is simply incorrect.
Response: After literature review, we came to know that sulphur contamination could be observed while XPS analysis, if sulphur related compounds are being analysed simultaneously, which could be the reason in our case, as there is gentamicin sulphate adjacent to polydopamine nanoparticles. We would consider it as an experimental error. It has been highlighted in manuscript.
Line 142 and elsewhere: anyone with experience in analytical spectroscopic methods knows that the absolute intensity values obtained on a sample depend on the sample thickness, surface roughness, etc. This is why researchers experienced in such methods never compare absolute values between samples. Instead, they compare component peak ratios in a given sample. For the authors to defend their comparison of absolute values between samples, by saying that others have also done so, seems, to me, to indicate that they do not understand the potential seriousness of what they have done.
Response: We do agree with the reviewer’s comments. Analysis has been repeated by comparing the percentage atomic concentration between samples.
Line 156: when a spectrum is analyzed, by using peaks of varying widths, that person is, in essence, fitting the spectrum with peaks at presupposed positions. That is, a bias is being imposed. The authors respond that they “haven’t observed any additional peak at constant fwhm values.”. I find this difficult to accept, for the following reason: by my rough estimation, the fwhm values in Figure 2(a) vary from about 1.5 eV for C-C, to about 2.5 eV for C-O/C-N. Had the authors fixed the fwhm value at about 1.5 eV, which is what is used in my lab, the three peaks they used could not possibly have been enough to fill the spectrum.
Response: The peak deconvolution has been repeated at their fixed ‘fwhm’ values by fixing ‘fwhm constraints’. A new figure has been replaced in the manuscript.
Line 172: the response to how the particle yield was calculated should have been included in the manuscript.
Response: The particle yield formula has been added in the manuscript.
Line 321: the authors indicate that the energy calibration was carried out by placing the C1s C-C peak at 284.8 eV. However, Figure 2(a) has it at 284.0 eV, Figure 2(b) has it at 284.7 eV, and Figure 2(c) has it at 284.6 eV.
Response: The energy calibration has been fixed to 284.8 eV while repeating peak deconvolution. The figure 2 has been replaced after calibration.
Round 3
Reviewer 2 Report
Review of Molecules-764501-Revision 2, “Polydopamine Nanosphere with In-situ Loaded Gentamicin and its Antimicrobial Activity” by R. Batul, M. Bhave, P. J. Mahon and A. Yu.
Although I have not changed my views on the value of this manuscript, the authors have made an effort to respond to my comments. It has clearly been a learning experience for them, and I hope they profit from it. I offer one further comment:
- The authors indicate that they calibrated the energy to the C-C C1s peak, placing it at 284.8 eV. However, the C-C peak in Figure 2a is placed at 284.0 eV, and that in Figure 2c, at 284.1 eV. Further, Table S1 places it at 284.1 eV. Although the authors do not mention it, and may be unaware of it, the energy resolution of monochromatic Al Kα X-rays is 0.45 eV. Thus, the difference between what is stated by the authors and what was actually used by them is real, and does not fall into the realm of experimental error. If nothing else, this is sloppiness.
Author Response
Comment
Although I have not changed my views on the value of this manuscript, the authors have made an effort to respond to my comments. It has clearly been a learning experience for them, and I hope they profit from it. I offer one further comment:
The authors indicate that they calibrated the energy to the C-C C1s peak, placing it at 284.8 eV. However, the C-C peak in Figure 2a is placed at 284.0 eV, and that in Figure 2c, at 284.1 eV. Further, Table S1 places it at 284.1 eV. Although the authors do not mention it, and may be unaware of it, the energy resolution of monochromatic Al Kα X-rays is 0.45 eV. Thus, the difference between what is stated by the authors and what was actually used by them is real and does not fall into the realm of experimental error. If nothing else, this is sloppiness.
Response
Thank you very much for the reviewer’s comment. We do appreciate your sharing with us your knowledge and expertise in XPS analysis. The peaks were re-calibrated using C1s peak at 284.8 eV, and the statement regarding the peaks calibration has been corrected in the manuscript. We hope the XPS analysis in the manuscript would be alright now. Thank you again for your valuable suggestions and time.